# Bone Morphogenetic Protein 2 (BMP-2) Aggregates Can be Solubilized by Albumin—Investigation of BMP-2 Aggregation by Light Scattering and Electrophoresis

**DOI:** 10.3390/pharmaceutics12121143

**Published:** 2020-11-25

**Authors:** Julius Sundermann, Holger Zagst, Judith Kuntsche, Hermann Wätzig, Heike Bunjes

**Affiliations:** 1Institut für Pharmazeutische Technologie und Biopharmazie, Technische Universität Braunschweig, D-38106 Braunschweig, Germany; j.sundermann@tu-braunschweig.de; 2Institut für Medizinische und Pharmazeutische Chemie, Technische Universität Braunschweig, D-38106 Braunschweig, Germany; h.zagst@tu-braunschweig.de (H.Z.); h.waetzig@tu-braunschweig.de (H.W.); 3Department of Physics, Chemistry and Pharmacy, University of Southern Denmark, DK-5000 Odense, Denmark; kuntsche@sdu.dk

**Keywords:** BMP-2, albumin, protein aggregation, protein-protein interactions, protein solubilization

## Abstract

Bone morphogenetic protein 2 (BMP-2) has a high tendency to aggregate at physiological pH and physiological ionic strength, which can complicate the development of growth factor delivery systems. The aggregation behavior in differently concentrated BMP-2 solutions was investigated using dynamic and static light scattering. It was found that at higher concentrations larger aggregates are formed, whose size decreases again with increasing dilution. A solubilizing effect and therefore less aggregation was observed upon the addition of albumin. Imaged capillary isoelectric focusing and the simulation of the surface charges of BMP-2 were used to find a possible explanation for the unusually low solubility of BMP-2 at physiological pH. In addition to hydrophobic interactions, attractive electrostatic interactions might be decisive in the aggregation of BMP-2 due to the particular distribution of surface charges. These results help to better understand the solubility behavior of BMP-2 and thus support future pharmaceutical research and the development of new strategies for the augmentation of bone healing.

## 1. Introduction

Bone morphogenetic proteins (BMPs) are members of the transforming growth factor-beta (TGF-β) family and are well known for their osteoinductive potential [1]. Extracellular signal transduction by the TGF-ß family proteins, which occurs on almost all metazoan cell types, has strict temporal and spatial requirements for appropriate signal levels. The regulation and transport of the proteins at the extracellular level is achieved by a complex interplay of several agonists and antagonists. Due to their very high binding affinity to BMP-2, the network of extracellular matrix proteins and proteoglycans can be regarded as a “sink” for BMP-2, which prevents its uncontrolled transport. However, the transport of BMP-2 to distant cells can be enabled by protein–protein binding with chordin, which leads to solubilization of the protein [2].

Due to their bone-inducing effect, recombinant human BMPs (rhBMPs) have been intensively investigated for bone healing and spinal fusion. As a result of this research, BMP-2 has received approval by the FDA in combination with a special medical delivery device (INFUSE^®^). The growth factor in this medical device-API-combination product is provided in a freeze-dried formulation that has to be reconstituted and applied to the device by the surgeon immediately before implantation [3]. In the marketed BMP-2 formulation, the basic disulfide-bridged homodimeric protein, which has an isoelectric point (pI) of 8.5 [4] and a size of 26 kDa [5], is buffered at pH 4.5 after reconstitution. At this pH, the protein has a strong positive charge and is considered to be largely soluble. However, Schwartz et al. demonstrated by size exclusion chromatography that the protein is partially aggregated in this buffer [6]. They also found that BMP-2 retains its full integrity in the precipitated state and is completely biologically active after redissolution. Further investigations by Luca et al. showed that the aggregate content in the marketed formulation might be even higher and that aggregate size and aggregation tendency of BMP-2 increase with increasing pH, which can be explained by a reduction of electrostatic repulsion when approaching the pI [7]. The formation of large BMP-2 aggregates (>1 µm) can be induced simply by increasing the pH from 4.5 to 6.5 in the marketed formulation INFUSE^®^ [7].

The solubility of the protein is also strongly influenced by ionic strength and buffer conditions, which lead to an extremely poor solubility of BMP-2 in Dulbecco’s phosphate buffered saline (PBS) at pH 7.4. The aggregation tendency is likely to be even more pronounced for *Escherichia coli (E. coli)*-derived BMP-2 than for BMP-2 from chinese hamster ovary (CHO) cells due to the lack of glycosylation (3–8 mannose residues at Asp^338^) and was investigated by Quaas et al. using dynamic light scattering and centrifugation-based precipitation experiments. In that study it was also shown that the precipitated BMP-2 aggregates can be redissolved by changing the buffer conditions and that BMP-2 within the aggregate particles retains its biological activity after redissolution [8].

The development of BMP-2-containing delivery systems necessitates in vitro release experiments at physiological pH. There are about a hundred published release studies with BMP-2 every year. In many cases, the release studies are carried out in PBS without any solubility increasing additives [9,10,11,12,13,14,15,16]. However, there are only a few publications that refer to the low solubility of the protein and the associated potential influence, both on the release itself (violation of sink conditions), and on the quantification of the partially aggregated protein [17]. Especially if analytical methods that are not robust against the influence of aggregation, such as ELISA are used for BMP-2 quantification, the results can be misleading.

A promising approach to improve the solubility of hydrophobic cytokines is the use of albumin [18]. The use of albumin as an additive allows a much more robust quantification of BMP-2 (manuscript in preparation). The addition of 0.1–1% albumin to the release buffer has been applied as a measure to prevent low BMP-2 recovery [19,20]. Loss of BMP-2 can also be partly explained by adsorption to surfaces [21]. However, surface adsorption can be minimized by using special low-binding vessels [22]. Reduced BMP-2 recoveries are therefore presumably mainly a result of aggregation. The aim of this study was to gain a deeper understanding of the BMP-2 aggregation process and to find out in which way it is influenced by the presence of albumin.

## 2. Materials and Methods

Unless otherwise stated all chemicals including human serum albumin (HSA) with a purity of ≥99% (cat. No. A3782) and bovine serum albumin (BSA) with a purity of ≥98% (cat. No. A7030) were purchased from Sigma-Aldrich. 2-(N-morpholino)ethanesulfonic acid (MES) with a purity of ≥ 99% was purchased from Carl Roth, Karlsruhe, Germany. *E. coli*-derived recombinant human bone morphogenetic protein 2 (rhBMP-2), simply referred to as BMP-2 in the following, was produced at the Institute for Technical Chemistry, Leibniz University Hannover, as previously described [8]. Unless otherwise stated BMP-2 was taken from aliquots with a concentration of 132 µg/mL in 50 mM MES buffer pH 5, stored at −80 °C and was thawed at room temperature before use.

### 2.1. Imaged Capillary Isoelectric Focusing (icIEF) of BMP-2

The imaged capillary isoelectric focusing (icIEF) experiments were performed with a Maurice system (ProteinSimple, San Jose, CA, USA), controlled by Compass for iCE 2.0.10, which was also used for data evaluation. The autosampler’s temperature was set to 10 °C. The separation took place in an icIEF cartridge containing a fluorocarbon-coated capillary with 100 µm internal diameter and 5 cm effective length. Sample was loaded for 55 s and then focused at 1.5 kV for 1 min, followed by a separation of 14 min at 3 kV. Detection was carried out by native fluorescence (ex. 280 nm, em. 320–450 nm) with 70 s exposure.

The reagents for operating the system were part of a cIEF Chemical Test Kit from ProteinSimple, San Jose, USA. Namely, 80 mM phosphoric acid in 0.1% methyl cellulose as the anolyte, 100 mM sodium hydroxide in 0.1% methyl cellulose as the catholyte, 0.5% methyl cellulose (MC) and fluorescence calibrations standard were used. The pI markers 4.05, 7.05, 9.50 and 9.99 and 1% MC solution were obtained from ProteinSimple, too. L-arginine was from Alfa Aesar, Kandel, Germany. Pharmalyte 3–10 was from GE Healthcare, Uppsala, Sweden.

First, stock solutions consisting of 10 M urea and 26.32 mg/mL HSA (solution A) or 10 M urea and 0.143% (*w*/*v*) Tween 20 in 1% MC (solution B) were prepared. Next, 1204 µL of solution A, 1109 µL of solution B, 31.7 µL of 500 mM L-arginine, 126.8 µL Pharmalyte 3–10 and 15.84 µL of each pI marker were mixed, vortexed and centrifuged at 10,000 rcf for 3 min. Ethylurea was then dissolved in this mix, leading to a concentration of 2.2 M in the final sample. Vortexing and centrifugation at 10,000 rcf for 3 min followed. Immediately before the measurement, 88 µL of the ethylurea-containing solution, 1.1 µL of BMP-2 solution (1 mg/mL in 1 mM HCl) and 20.8 µL of water were carefully mixed in a 96-well plate. Sample injections were performed in triplicate.

### 2.2. 2D Blue Native/SDS-PAGE, Anodic Native and Cathodic Native PAGE

Blue native PAGE was carried out as described by Fiala et al. [23]. Coomassie Brilliant Blue (G-250) was purchased from Bio-Rad. Any kD Mini-PROTEAN TGX Precast Gels (Bio-Rad) were used for both dimensions. Lanes of first dimension gels were cut out and equilibrated in non-reducing Laemmli buffer (Bio-Rad) for 30 min. After incubation, the gel strips were applied to a second-dimension gel with cut-off loading pockets for the subsequent SDS-PAGE (Figure 1). SDS-PAGE and anodic native PAGE were performed according to the manufacturer’s instructions (Bio-Rad Mini-PROTEAN protocol). Cathodic native PAGE and gel casting was carried out as described by Säftel et al. [24].

Proteins were visualized by staining the gels with the Bio-Rad Silver Stain Plus kit or QC Colloidal Coomassie Stain according to the manufacturer’s instructions.

### 2.3. Western Blot

After native PAGE, proteins were electroblotted to a 0.45 µm PVDF membrane (Immobilon-P, Cat. IPVH000010, Merck) using the Mini Trans-Blot Module (Bio-Rad) according to the manufacturer’s instructions. BMP-2 was detected using the Pierce SuperSignal West Femto kit (Cat. 35081, Thermo Fischer Scientific, Waltham, MA, USA) with a rabbit anti-BMP-2 primary antibody (Cat. 500-P195, PeproTech, Rocky Hill, NJ, USA).

### 2.4. Asymmetric Flow Field-Flow Fractionation with Multi-Angle Light Scattering (AF4/MALS)

Of BMP-2 0.904 mg (freeze-dried in 4 mL 50 mM MES buffer pH 5) were reconstituted with 2 mL of 50 mM MES buffer, pH 5, resulting in a stock solution of 0.452 mg/mL of BMP-2. This BMP-2 stock solution was stored at 4–8 °C. For reference, 10 mg of human serum albumin (HSA, A3782 Sigma-Aldrich, purity of ≥99%) were dissolved in 1 mL of 50 mM MES buffer. After equilibration at room temperature for 30 min, the solutions were further diluted to the concentrations indicated in Table 1.

The AF4 system consisting of an isocratic pump and degasser, autosampler (all from Agilent) and an Eclipse 3+ instrument (Wyatt Technology Europe, Dernbach, Germany) was connected to a multiangle light scattering (DAWN Heleos II, Wyatt Technology) and a differential refractive index (dRI, Optilab rEX, Wyatt Technology) detector. The AF4 channel (long channel, Wyatt Technology) was equipped with a membrane of regenerated cellulose (MWCO 10 kDa).

Sample volumes of 20, 40, 60, 80 and 100 µL sample were subsequently injected over 2 min into the AF4 channel in the focus mode (2 mL/min focus flow and 0.2 mL/min injection flow). The samples were further focused for 1 min and then eluted with a channel flow of 1 mL/min without applied cross flow. Blank injections (buffer) were used for baseline correction.

Data analysis was done with the Astra software version 7 (Wyatt Technology) using the dRI signals as concentration source (dn/dc = 0.185) and applying the reciprocal Zimm method (Debye fitting). The average molecular weight (M_w_) over the defined peak area was determined using the Zimm Equation (1): (1)$$M_{w}\;=\;{\left(\frac{Kc}{R_{0}\;}{\;-\;2A}_{2}c\right)}^{-1}$$
with c being the sample concentration, M_w_ the weight average molar mass, K an optical constant, R0 the Rayleigh ratio extrapolated to zero scattering angle and A2 the second viral coefficient. For molar mass calculations, A2 was set to 0 in this study, as the A2 could not be determined.

Molar mass distributions by weight or number can be calculated according to Equations (2) and (3):(2)$$M_{w}=\;\frac{\sum^{\;}\left(c_{i}M_{i}\right)}{\sum^{\;}c_{i}}$$
(3)$$M_{n}=\;\frac{\sum^{\;}c_{i}}{\sum^{\;}\frac{c_{i}}{M_{i}}}$$

The average concentrations of protein in the peaks upon sample elution in AF4 were calculated by dividing the detected mass by the investigated volume. A constant investigated volume of 4 mL was defined by setting a peak width of 4 s.

### 2.5. Dynamic Light Scattering (DLS) and Nanoparticle Tracking Analysis (NTA) Measurements

The intensity weighted size distribution of BMP-2 aggregates in different media was determined by dynamic light scattering (DLS; Zetasizer Nano ZS, Malvern Panalytical, Worcestershire, United Kingdom) at BMP-2 concentrations between 13.2 and 132 μg/mL. Light scattering was measured in a low-volume quartz cuvette (Ultra-Micro Cell ZEN2112, Malvern) at 25 °C and an angle of 173° for 300 s after an equilibration time of 300 s.

The DLS size distribution was calculated from the autocorrelation curves applying the protein analysis algorithm (non-negative least squares analysis followed by L-curve algorithm using 300 size classes) using the Zetasizer Software 7.11 (Malvern). The lower threshold was set to 0.00 (default: 0.05) and the lower limit of the display range was set to 1 nm (default: 0.01 nm). The volume size distributions were calculated by Mie theory assuming a refractive index (RI) of 1.450 and a particle absorption of 0.001.

The mean number size distribution of BMP-2 aggregates in PBS pH 7.2 and MES pH 5 at a BMP-2 concentration of 13.2 μg/mL was measured by NTA (NanoSight NS300, Malvern) at 20 °C. For each measurement, five 1-min videos with the camera level set to 10 were captured. After capture, the videos were analyzed by the in-built NanoSight software version 3.2 with a detection threshold of 10.

### 2.6. Computation of BMP-2 Isoelectric Point and Electrostatic Potential Map

Atomic coordinates of the crystallographic structure of the native BMP-2 homo dimer were extracted from the BMP-2-BMP receptor IA complex (PDB code: 1REW) [25]. pI and the protonation state of BMP-2 were evaluated by means of H++ software [26]. The electrostatic surface potential was computed using the Adaptive Poisson-Boltzmann Solver (APBS) module [27] within the Python Molecule Viewer [28].

## 3. Results

### 3.1. Imaged Capillary Isoelectric Focusing (icIEF) of BMP-2

The icIEF of BMP-2 in urea solutions of different concentrations led to several, non-reproducible peaks in the pH range between pH 7 and pH 8 (Supporting Information, Appendix A). As BMP-2 is known to have the ability to renature after chaotropic denaturation [8], it was concluded that urea solutions were not chaotropic enough to provide a substantial BMP-2 solubilization during iciEF. For this reason, the stronger denaturizing agent ethylurea was applied.

The electropherograms of BMP-2 solutions in 2.2 M ethylurea displayed two reproducible peaks at pH 8.2 and pH 8.5 (Figure 2). Based on observations of Geravais and King, who explored the use of ethylurea for “difficult-to-denature proteins” in icIEF before [29], it might be possible that two peaks represent the same molecule, only in different states of denaturation. Since both peaks represented about 50% of the total peak area independent of the ethylurea concentration (2–4 M, data not shown), it was concluded that the BMP-2 was fully denaturized, and that indeed two different charge variants of BMP-2 were present in the tested samples.

### 3.2. Theoretical Isoelectric Point and Electrostatic Potential Map of BMP-2

Based on investigations by Li et al. [30] a value of 10 for the internal dielectric constant ε_i_ of the protein was estimated. Due to the small size of the protein, the IP calculation was additionally performed assuming an internal dielectric constant of 80. At a salinity of 0.15 M, the calculated IP of a BMP-2 monomer (PDB code: 3BMP) was 6.08 assuming an ε_i_ of 10 and 6.47 assuming an ε_i_ of 80. The calculated IP of a native BMP-2 homo dimer (extracted from PDB code 1REW) was 5.45 assuming an ε_i_ of 10 and 5.62 assuming an ε_i_ of 80. The estimated IP values depend on the assumed salinity as well.

The electrostatic potential map of BMP-2 displays a tripolar charge distribution (Figure 3), which has been visualized previously [5]. Almost all of the positive charge is concentrated in the central area, while the negative charge is located at the two tips of the protein. Between these regions and on the back, there are distinct hydrophobic patches. Two BMP-2 molecules can approach each other in such a way that the charged and hydrophobic regions are completely complementary to each other (Figure 3B,C).

### 3.3. SDS-PAGE, Blue Native-PAGE and 2D Blue Native/SDS-PAGE of BMP-2 and BSA

No signs of impurities or oligomerization were visible in conventional nonreducing SDS-PAGEs of BMP-2 (Figure 4). The apparent molecular weight of BMP-2 in SDS-PAGE (23.3 kDa) is about 10% less than the formula weight of 26 kDa. This type of deviation is known as “gel shifting” and has been observed for other hydrophobic proteins like membrane proteins because of an increased SDS binding [31].

Western blots of blue native PAGEs of BMP-2 solutions show a broad size distribution with two detectable maxima at approximately 60 and 100 kDa, which refer to dimers and tetramers of the 26 kDa BMP-2 dimer (Figure 5C). BSA or HSA (including contained serum proteins with sizes between 100 and 250 kDa) were applied as reference in all electrophoretic investigations shown. As can be seen in lane VI of Figure 5C, the high amounts of BSA applied together with BMP-2 blocked the PVDF membrane during blotting, resulting in a negative staining in the 66 kDa BSA region.

A wide distribution of BMP-2 aggregate sizes was also visualized by the silver staining of the blue native PAGE (Figure 5A,B). The silver staining revealed that most of the BMP-2 had actually just entered a few millimeters into the collection gel, which indicates aggregates being larger than 250 kDa. Those large BMP-2 aggregates were detected in the Western blot with a much lower sensitivity. It appears that they either had a lower affinity to the detection antibodies or that they could not quantitatively be blotted onto the PVDF membrane because of the technical limitations of the Western blot technique for the transfer of proteins with a high molecular weight above about 200 kDa [32]. Interestingly, considerably less of both large and small BMP-2 aggregates were detectable by silver staining when BMP-2 was applied together with albumin. Nevertheless, there were still no 26 kDa BMP-2 dimers present.

The regions of the blue native PAGEs marked with the blue boxes in Figure 5B were cut out of an identical, but unstained gel and were applied to three SDS-PAGEs (Figure 5D).

As expected, albumin and the larger serum proteins were found on a diagonal line (highlighted by dashed yellow lines) because they had been separated twice by their different protein sizes (lane I and IV in Figure 5D). BMP-2 was found on a curved horizontal front because it had been separated once by the various aggregate sizes and once by its uniform protein size (lane III in Figure 5D, see Figure 1 for further clarification). The reason for the curvature of the line on which BMP-2 was distributed is probably the asymmetrical structure of the precast gels used. As long as the proteins were still migrating through the cut-out parts of the first gels, the pore size of the polyacrylamide gel differed depending on how far they had advanced. The less the proteins advanced during the first dimensions, the larger the pores of the initial PA gel and the further they migrated in the second dimension.

The second dimension of lane III in Figure 5 revealed that the BMP-2 in the mixture with BSA was almost completely localized within the BSA band of the blue native PAGE.

During anodic native PAGE at pH 8.3, BMP-2 did not migrate into the gel (sample buffer pH 6.8, running buffer pH 8.3; Figure 6A). However, if BMP-2 was applied in the mixture with BSA or HSA, it could be detected in the pockets of the gel by Western blotting.

In order to prevent the protein from escaping the gel, it was enclosed in the middle of a precast gel before the run. To achieve this, the protein was applied into drilled holes in the middle of the gel. In this setup, the BMP-2 aggregates moved towards the anode during native electrophoresis at pH 8.3, indicating that the BMP-2 aggregates were negatively charged at this pH (Figure 7).

After cathodic native electrophoresis at pH 7.6, BMP-2 aggregates could be detected in the gel by silver staining (Figure 6B), indicating that they were positively charged at this pH. Therefore, the pI of the BMP-2 aggregates is between 7.6 and 8.3 according to these measurements. This result does not contradict the cIEF results from Section 3.1. Isoelectric points can easily vary by 1 unit depending on the experimental conditions, e.g., related to interactions with ions from the solution [33].

Albumin did not migrate in the direction of the cathode under these conditions and was therefore not found in the gel. If BMP-2 was applied in the mixture with albumin, it also no longer migrated into the gel, but was probably transported with the BSA towards the anode.

### 3.4. Recovery and Molar Mass Determinations (AF4/MALS) in MES Buffer pH 5 and PBS pH 7.4

BMP-2 solutions in MES buffer (0.452 mg/mL and 0.226 mg/mL) were subsequently injected into MES buffer pH 5 and PBS pH 7.4. The recovery in MES buffer was between 25 and 90% in dependence on injected mass (increased recovery with increasing injected mass) of the stock solution (Figure 8).

No signals deviating from the blank value were detectable upon BMP-2 injection into PBS, and the recovery of BMP-2 in PBS pH 7.4 was therefore 0% (Figure 8). As a reference, the recovery of HSA in PBS was 95.4% ± 2.2% (*n* = 10).

The determined molar masses of the different injection series of BMP-2 into MES buffer are presented in Figure 9. In general, the molar masses increased with increasing injected mass (e.g., series a, b and c). Injection series a, which was applied in the time period from 22 to 85 min after reconstitution, showed lower molar masses in the first two injections (22 and 37 min after reconstitution), than injection series b (302–365 min after reconstitution) and c (436–500 min after reconstitution). Due to the sequential nature of the injections, there may also be an effect related to the different sample storage periods after reconstitution.

The concentration-correlated M_w_ increase appeared to be steeper for the injection series c, which had a lower stock concentration compared to a and b. The measured molecular weight seemed therefore to depend not only on the measured concentration and storage time of stock solution, but also on the initial concentration before injection. For this reason, the molecular weight was examined in relation to the degree of dilution (ratio of determined concentration in AF4 and concentration of stock solution, Figure 9B). The determined M_w_ exponentially decreased with the degree of dilution, for both tested initial concentrations. The low molecular masses of 45 kDa and 310 kDa determined for the very first injections (marked in red) appeared to be out of line with the trend shown.

The recovery of BMP-2 after injection into MES buffer was 62.7% ± 19.6% (*n* = 10).

To check whether BSA may decrease the aggregation tendency of BMP-2, HSA stock solution was added to the BMP stock solution (both in MES buffer) in order to reach concentrations of BMP-2 and HSA of 0.20 and 1.15 mg/mL, respectively. This corresponded to molar concentrations of 7.7 µmol/mL BMP-2 and 15.1 µmol/mL HSA. As a control, HSA solution in MES (1 mg/mL) was used. Two injection series each into PBS buffer were carried out (Figure 10). Between each of the two measurement series, the proteins in MES pH 5 were stored at 4–8 °C for 24 h. The mean value for the determined weight average-molecular weight of the in total 10 injections of HSA alone was 62.7 ± 2.2 kDa (M_w_/M_n_ = 1.025 ± 0.012), i.e., approximately equivalent to the expected molecular weight of 66.4 kDa, and showed no concentration or overnight storage correlated alteration. For the HSA/BMP-2 mixtures, molar masses and polydispersity (M_w_/M_n_) increased with increasing injected mass, e.g., lower dilutions from 1.027 to 1.099 (c). The concentration-correlated increase of the M_w_, highlighted by the dotted line in Figure 10, and the increase of polydispersity was more prominent for the first injection series.

### 3.5. DLS and NTA Measurements of BMP-2 in MES Buffer pH 5 and in PBS pH 7.2

According to DLS measurements, 96% of BMP-2 in MES buffer pH 5 was in native dimeric or oligomeric form with a mode diameter of 7.8 ± 0.7 nm (SD) and 3% was apparently in the form of larger aggregates with various sizes and mode diameter of 136 ± 239 nm (SD; Figure 11C). It is important to note that, despite the high quantitative proportion of small particles (<10 nm) detected, this fraction of the particle size distribution might not be detected in the standard configuration of common DLS software. Since the scattered light intensity of these small particles is less than 5% of the total intensity, it is cut out before the evaluation in the standard configuration of the “protein analysis” model of the Malvern Zeta Sizer software version 7.11 (Supporting Information, Section 2). This selective evaluation was avoided by setting the “lower threshold” to 0.00 (standard: 0.05). As a result of this adjustment not only the dynamic light scattering of monomeric BMP-2, but also dynamic light scattering arising from even smaller structures that might be attributed to MES buffer molecules (peak at 0.7 ± 0.04 nm) was included in the calculated distribution (Supporting Information, icIEF measurements using urea, Appendix A). To avoid the size distribution of the BMP-2 aggregates being biased by these small structures, the lower limit of the display range was set to 1 nm (default: 0.01 nm).

Using NTA, which only detects aggregates above a size of approximately 20 nm, the same BMP-2 solutions showed a mode number-weighted diameter of 165 nm (Figure 12C). Aggregates in the size range of approximately 20–360 nm also prevailed in the intensity distribution of the DLS measurements, which was independent of the above-mentioned software configuration (Figure 11E). In PBS at pH 7.2 large BMP-2 aggregates in various sizes with sedimentation tendency impeded precise size determination by DLS and NTA. The DLS measurements showed an increase in correlation at high decay times and elevated baselines, both indicating number variations and non-random particle motion due to sedimentation [34] (Figure 11B). These factors indicate aggregate particles larger than about 5 µm, which no longer exhibited sufficient Brownian motion for the DLS theory to be applied correctly. DLS intensity and volume size distributions calculated from these correlation functions (Figure 11D,F) must therefore be examined from a critical perspective. Due to the low solubility of BMP-2 in PBS, DLS measurements with higher concentrations could not be performed. The measurement of lower concentrated BMP-2 solutions in PBS could not be evaluated due to low light scattering intensity.

In NTA measurements, the scattered light from large aggregates, which as a whole showed no trackable Brownian motion, was optically visible (Figure 12B). Instead of tracking actual aggregate particles like in MES buffer pH 5 (Figure 12), the NTA software could only track fluctuations within those particles. Therefore the calculated results did not reflect the actual particle size (data not shown).

## 4. Discussion

### 4.1. Isoelectric Point and Charge

The pI of denatured proteins can be predicted by models that use sets of empirically determined pKa values for all titratable sites [35]. Two exemplary calculation bases are the pKa values determined by Bjellqvist et al. (ExPASy) [36] and Halligan et al. (ProMoST) [37]. The calculation of the pI of the BMP-2 dimer sequence (2 × PDB code: 3BMP), using the web-based program Prot pi [38], yields a predicted pI of 8.7 (ProMoST) and 8.9 (ExPASy). However, these calculations take into account only the amino acid sequence and not the protein structure.

By icIEF a charge heterogeneity with two BMP-2 species (pI 8.2 and pI 8.5) with approximately equal amounts was observed. With reference to Uludag et al. [39] the pI of both, CHO cell-derived BMP-2 and *E. coli*-derived BMP-2, is commonly cited as 8.5. However, according to that source, the pI of CHO BMP-2 is “a doublet with the pI of ~9” and the pI of *E. coli* BMP-2 is given as approximately 8.5. By plotting the grey value of the IEF gel depicted in the publication of Uludag et al. relative to the depicted IEF scale, it can be seen that the band of *E. coli* BMP-2 could also be considered as two bands at a distance of about 0.3 pI units, which is fully consistent with the results presented in this study (Supporting Information, Appendix A).

The presence of two peaks/bands in IEF can indicate the presence of acidic or basic charge variants. Possible causes for acidic charge variants in the BMP-2 samples are deamidation, non-classical disulfide linkage and reduced disulfide bonds. On the other hand, basic charge variants could be caused by the isomerization of Asp, Met oxidation, amidation, incomplete disulfide bonds, fragments or aggregates. Since previous investigations have shown that the disulfide bridges of the BMP-2 used in this study were correctly formed and because the protein was not exposed to any oxidative or reductive stress, most of the above listed possible causes are extremely unlikely [40]. However, at the acidic buffer conditions, which have to be applied in order to achieve a sufficient BMP-2 solubility, aspartic acid (Asp) residues are susceptible to form cyclic imide intermediates (succinimide). The succinimide ring hydrolyzes into Asp and isoaspartic acid (isoAsp) at an Asp to isoAsp ratio of 1:3 (Asp isomerization) [41]. Glycosylated BMP-2 contains at least one Asp residue in each monomer (Asp335), which is very prone to Asp isomerization at slightly acidic pH. Schwartz et al. found that, after a 25 °C storage of BMP-2 at pH 4.5 (identical buffer composition as used by Uludag et al.) for 5 months, 68.8% of Asp335 was isomerized [6,42]. Due to the lack of glycosylation on Asp338, *E. coli*-derived BMP-2 has a second residue, which possibly is susceptible for Asp isomerization. Asp isomerization leads to basic protein species, which could explain the doublet peak [43]. However, the slightly acidic conditions that have to be applied for BMP-2 handling might also have caused a succinimide-mediated deamidation of Asparagine (Asn) [41]. Asn deamidation results in the conversion of Asn to Asp whereby a negative charge is introduced, which leads to acidic protein species [43].

The pI of proteins in native and denatured state may differ significantly. For BSA, for example, there is a difference of one pH unit [44]. The pI determination is also influenced by high urea concentrations [45]. Due to its extremely poor solubility, native BMP-2 could not successfully be measured using icIEF. To our knowledge, there is no published experimental data on the pI of native BMP-2. Due to the migrating direction of BMP-2 towards the cathode during cathodic native PAGE at pH 7.6 and the migrating direction towards the anode in the anodic native PAGE at pH 8.3, the pI of undenatured BMP-2 can be limited to the range of 7.7–8.3. However, it must be taken into account that the BMP-2 under investigation was completely aggregated at this pH and that aggregation can shift the pI towards higher values [43]. The determined pI of 7.7–8.3, which was derived from the electrophoretic properties under native conditions therefore only applies to aggregates of native BMP-2.

Interestingly, the pI computed for native BMP-2 by H++ was quite low (pI = 6.1), which may be attributable to the special charge distribution of the protein. The prediction of a protein’s pI is based on the summation of its partial charges, calculated for each ionizable group using the Henderson–Hasselbalch equation at different pH values [46]. The decisive factor is the determination of the different pKa values of the corresponding titratable groups. By calculating the pKa values of the ionizable groups of BMP-2 with the H++ software, site interactions between the positive center and the negative tips of the protein were taken into account [26]. The shielding of the central positive charges by the distal negative charges (Figure 3) may explain why the calculated and the experimentally determined effective pI of a native BMP-2 dimer are considerably lower than the calculated pI of the denatured protein.

### 4.2. Aggregation

Despite the very low concentrations (<15 µg/mL) in the AF4/MALS experiments, there was obviously a strong attractive interaction of the single BMP-2 dimers, which resulted in the aggregation observed. The strength of attractive or repulsive interactions of proteins can generally be assessed by determining the second virial coefficient (A2) using static light scattering. However, a determination of the A2 value using the Zimm plot would have required the monomeric form of the molecules at least at higher dilutions. Since this condition could not be met for BMP-2 within the measurable concentration range, a determination of A2 using static light scattering turned out to be not feasible.

Although BMP-2 is commonly regarded soluble under these conditions, aggregates could be detected in MES buffer pH 5 by dynamic and static light scattering and by NTA. There was a broad particle size distribution between about 7 and 300 nm. Since the aggregated particles did not exceed a size that would lead to sedimentation in centrifugation-based solubility testing, they could still be considered “soluble” under these conditions. In PBS buffer pH 7.4 DLS and NTA measurements indicated the formation of large aggregates with sedimentation tendency. However, it was not possible to precisely measure the aggregate particle size in PBS with DLS or NTA because the aggregate particles were too large and the particle size distribution too wide for an accurate measurement. In addition, NTA partly tracked movements within the aggregates rather than the movement of whole aggregate particles.

In the AF4/MALS measurements it was observed that the size of the BMP-2 aggregates in MES buffer pH 5 varied depending on concentration. As the concentration in the stock solution was much higher than during the individual measurements, the sample was diluted to a different extent during each injection. In effect, it was not tested in the experiment how the aggregates grew in dependence on the concentration, but how the aggregates shrank depending on the conditions of different dilution. Therefore, one may conclude that the concentration-dependent aggregation process was reversible. In MES buffer pH 5 the different aggregate sizes of BMP-2 appear to be in a concentration-dependent equilibrium (Figure 13).

A possible cause for BMP-2 aggregation is a crosslinking of BMP-2 dimers via phosphate ions. This hypothesis was originally postulated by Abbatiello et al. [47]. Aggregation was also highly elevated in phosphate buffers in this study, however, a strong aggregation of BMP-2 also occurred without the presence of divalent ions.

Therefore, the complementarity of charges and hydrophobic regions illustrated in Figure 3 may be considered as an alternative possible cause for the extremely strong aggregation tendency of BMP-2. During cIEF measurements, BMP-2 precipitation occurred despite extremely high concentrations of the chaotropic osmolyte urea. Urea apparently had very little impact on the solubility of BMP-2. The addition of chaotropic osmolytes reduces the free surface energy of hydrophobic protein regions. In highly concentrated urea solutions, hydrophobic interactions are therefore largely eliminated [48]. The BMP-2 aggregation observed under these conditions suggests that the aggregation process is mainly driven by ionic interactions.

The ability of BMP-2 to interact with other positively charged molecules via ionic attraction despite its overall positive total charge has already been demonstrated for chitosan using molecular dynamics simulation [49]. Presumably, BMP-2 can enter into an ionic interaction with itself in the presented manner (Figure 3). In the arrangement shown, two ionic and one hydrophobic region between two BMP-2 molecules are complementary to each other. Obviously, simpler constellations are also conceivable, in which only one negative tip interacts with the positive center. Electrostatic attractions between BMP-2 molecules might explain the unusual salting in effect [47], according to which the solubility of BMP-2 increases with higher electrostatic shielding. The charge heterogeneity, which was detected by icIEF (cf. 4.1), might be an additional possible cause for the aggregation tendency.

### 4.3. BMP 2 Solubilization by Albumin

BMP-2 in 2D blue native/SDS-PAGE (Figure 5) and in cathodic and anodic native PAGE (Figure 6) exhibited a different electrophoretic behavior depending on whether it was applied with or without albumin. The band of BMP-2 aggregates was significantly reduced in blue-native PAGE when BMP-2 was applied together with albumin (Figure 5, blue boxes). However, BMP-2 dimers with a size of 26 kDa were in no case detectable under native conditions. In the second dimension it could be seen that BMP-2 was mainly located in the albumin band during blue native PAGE, which is a strong indication for a protein–protein interaction.

In cathodic native PAGE (Figure 6), the positively charged BMP-2 aggregates at pH 7.6 did not migrate into the gel when applied together with albumin, whereas BMP-2 aggregates alone were pulled into the gel under these conditions. This suggests that BMP-2 formed a negatively charged complex with albumin that was attracted towards the anode. In anodic native PAGE, BMP-2 aggregates could be detected inside the pockets of the gel by Western blot only if they were applied together with albumin (Figure 6).

According to precipitation experiments by Quaas et al., BMP-2 has to be considered as completely insoluble in PBS at pH 7.4. During AF4/MALS measurements, BMP-2 aggregates in PBS pH 7.4 did not reach the detectors, which was presumably a result of protein loss due to precipitation. However, when BMP-2 was applied in combination with HSA it could be detected indirectly by the concentration dependent increase of the determined molecular weights. The concentration dependent increase in molecular weight at the fixed HSA to BMP-2 ratio of about 2 to 1 indicates that aggregate sizes were still increasing with higher overall protein concentrations. This suggests either the presence of growing BMP-2 aggregates solubilized by bound HSA or that HSA itself was incorporated into the aggregates. Since this effect was less pronounced after one day of storage of the BMP-2 + HSA mixture in MES pH 5, it can be assumed that the formed BMP-2-albumin complexes tend to precipitate at pH 5. The reason for this could be the relatively poor solubility of HSA at pH 5, which may not be sufficient for solubilization of bound BMP-2.

## 5. Conclusions

Albumin improved BMP-2 solubility and influenced the electrophoretic behavior of BMP-2. For albumin, a chaperone-like protective effect against protein aggregation has been described [50]. However, since the aggregation of BMP-2 is based on low solubility and not on denaturation, other mechanisms are relevant in this case. The electrophoretic results indicate protein–protein interactions between albumin and BMP-2. Protein–protein interaction and protein aggregation are associated processes, which are strongly influenced, e.g., by pH value and “salting in” effects. The pI of native but aggregated BMP-2 appeared to be between 7.6 and 8.3, which is considerably lower than that of denatured BMP-2.

BMP-2 has a high affinity to positive, negative and hydrophobic surfaces [51,52]. Surfaces with such characteristics can be found on the protein itself. In solution, BMP-2 apparently tended to interact with itself, which resulted in the formation of aggregates even under the favorable conditions of 50 mM MES buffer pH 5. The self-interaction of BMP-2 dimers may be explained by the described complementarity of charged and hydrophobic regions. At pH 7.4, BMP-2 self-affinity led to the formation of insoluble (precipitating) aggregates. However, the binding affinity to added albumin seemed to be high enough to prevent BMP-2 self-interaction to some extent. The binding of BMP-2 to albumin might be driven by hydrophobic and by ionic interactions similar to the discussed BMP-2 self-interaction. BMP-2 and albumin have an opposite net charge at physiological pH and there are hydrophobic patches on the surface of both proteins [51,53]. Albumin is well known for its ability to bind proteins, especially if they have a high hydrophobicity, like apolipoproteins [8]. Due to the ubiquitous presence of albumin within the interstitial fluid, the binding of BMP-2 to albumin might play an important role for its interstitial mobility. An interaction between albumin and BMP-2 is therefore not surprising, even though, to our knowledge, it has not been described previously.

## Figures and Tables

**Figure 1 pharmaceutics-12-01143-f001:**
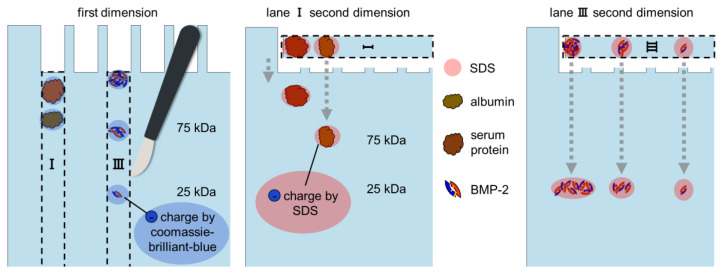
Illustration of the principle of a 2D Blue Native/SDS-PAGE showing the separation of BMP-2 aggregates and the separation of BSA (serum fraction V).

**Figure 2 pharmaceutics-12-01143-f002:**
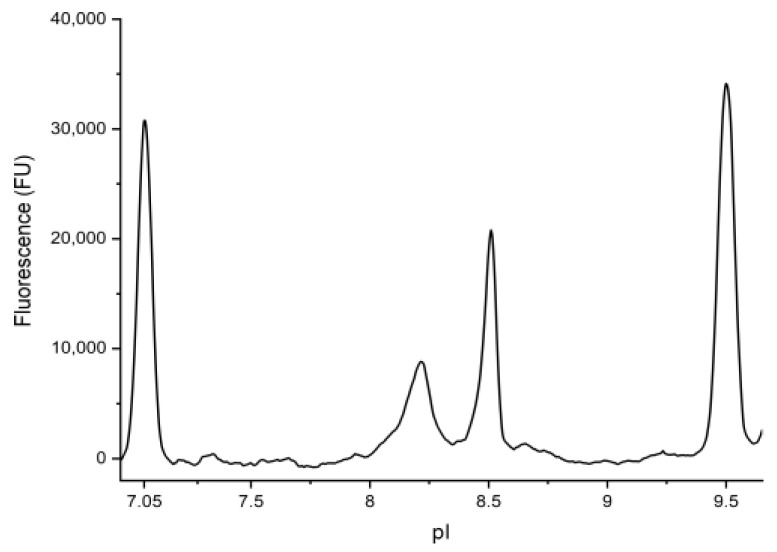
Representative electropherogram of BMP-2 (two peaks at 8.2 and 8.5) and adjacent isoelectric point (pI) markers (7.05 and 9.5).

**Figure 3 pharmaceutics-12-01143-f003:**
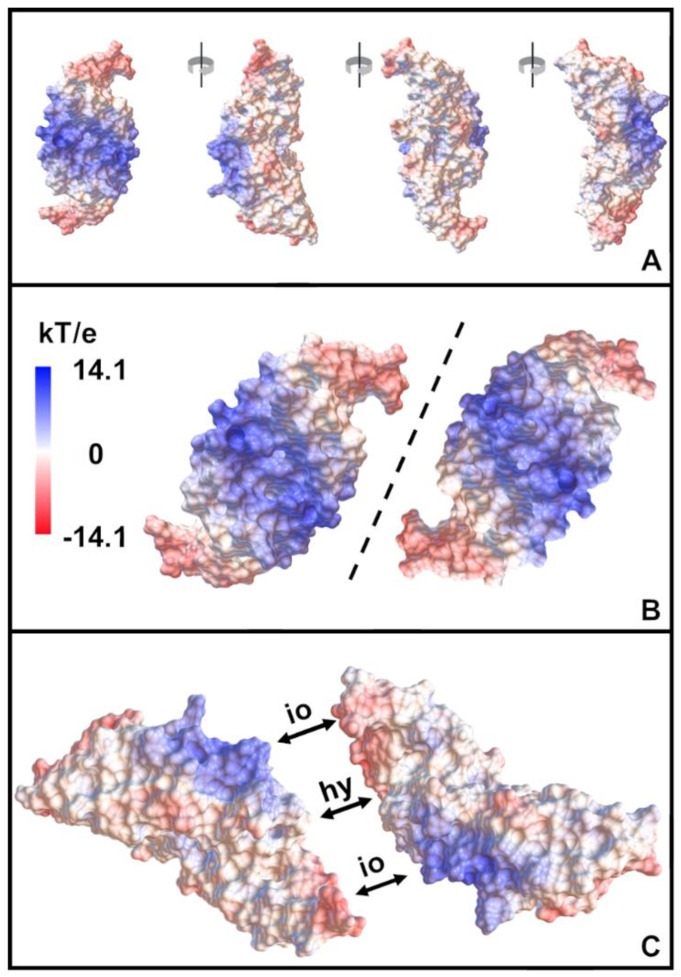
Electrostatic surface potential of BMP-2 at pH 5 as determined by continuum electrostatic calculations. (**A**): BMP-2 displayed from all sides and (**B**): view on the central cavities of two BMP-2 molecules showing the possible interaction patches frontally. The dotted line indicates the axis at which negatively charged (red) areas are reflected in complementary positively charged (blue) areas and at which hydrophobic (white) areas are reflected in other hydrophobic areas. (**C**): Possible interaction patches facing each other. Compared to (**B**), the two BMP-2 dimers were rotated 90° towards each other. io = possible electrostatic attraction, hy = possible hydrophobic interaction.

**Figure 4 pharmaceutics-12-01143-f004:**
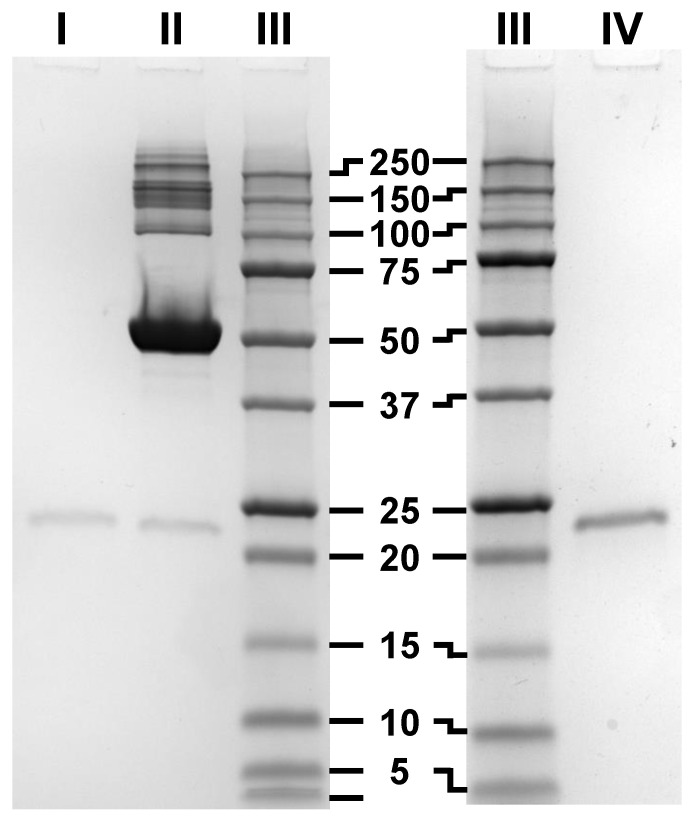
Representative Coomassie brilliant blue stained SDS-PAGE. I = 125 ng BMP-2, II = I + 30 µg BSA, III = Ladder, IV = 500 ng BMP-2.

**Figure 5 pharmaceutics-12-01143-f005:**
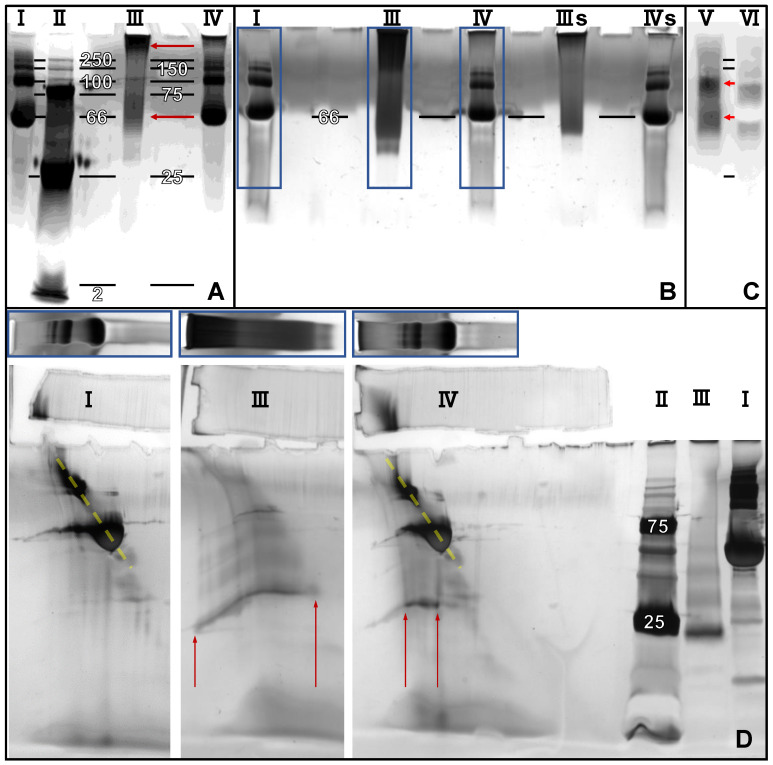
(**A**,**B**) silver-stain and (**C**) Western blot of blue native PAGEs. (**D**) Silver-stain of three second dimension gels of 2D Blue Native/SDS-PAGEs. Darker grey value corresponds to higher degree of light attenuation during transillumination (**A**,**B**,**D**) or to higher relative luminescence (**C**). Blue boxes indicate where the lanes were cut out of the blue native PAGE and how they were applied on the second dimension SDS-PAGEs. White labels indicate approximate molecular mass (kDa). Red arrows highlight distribution of BMP-2. I = 30 µg BSA, II = Ladder, III = 2 µg BMP-2, IIIs = 0.5 µg BMP-2, IV = I + III, IVs = I + IIIs, V = 16 ng BMP-2, VI = V + 10 µg BSA.

**Figure 6 pharmaceutics-12-01143-f006:**
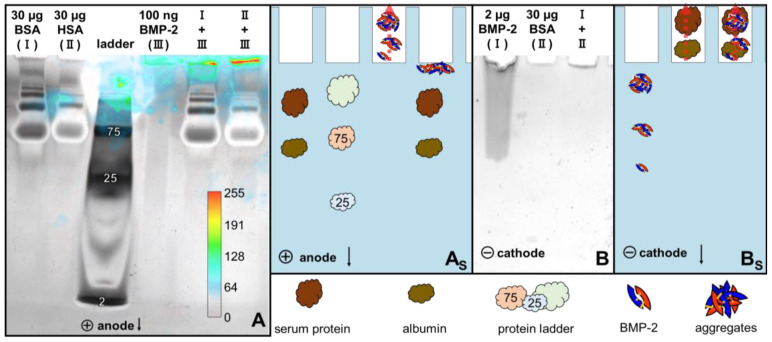
(**A**) Silver staining and overlay of luminescence intensity after Western blot of identical gel, after anodic native PAGE at pH 8.3, (**A_S_**): explanatory illustration of (**A**), (**B**) silver staining after cathodic native PAGE at pH 7.6, (**B_S_**): explanatory illustration of (**B**). Red arrows illustrate the apparent direction of movement.

**Figure 7 pharmaceutics-12-01143-f007:**
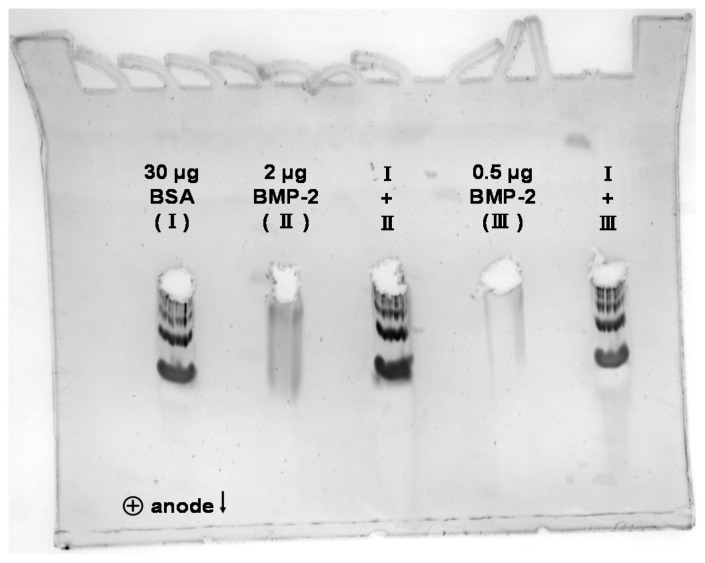
Silver staining after anodic native PAGE at pH 8.3 with proteins starting from drilled holes in the middle of the polyacrylamide gel.

**Figure 8 pharmaceutics-12-01143-f008:**
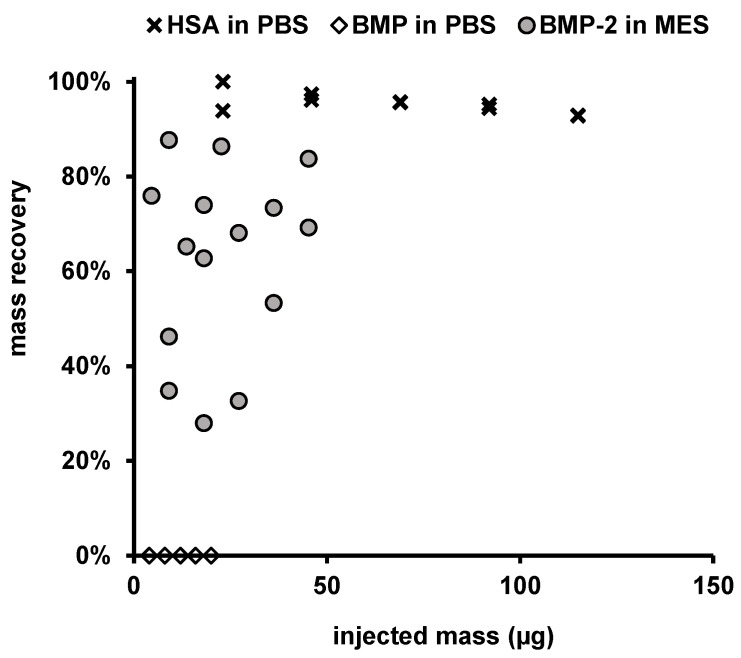
HSA and BMP-2 recovery (dRI detector of AF4 system) for different injected masses.

**Figure 9 pharmaceutics-12-01143-f009:**
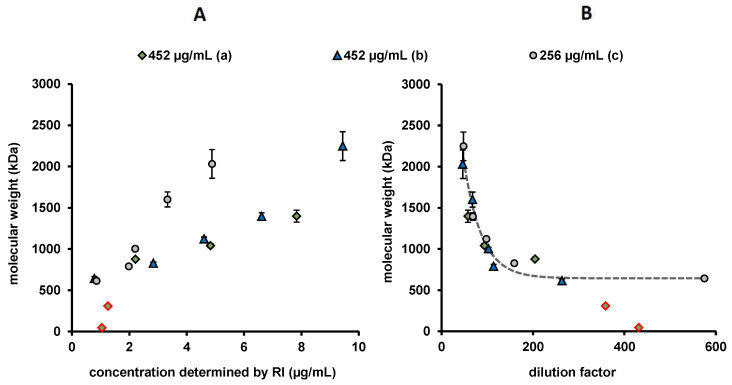
(**A**) Weight-average molecular weights (M_w_) of BMP-2 injection series a, b and c into MES buffer, determined by MALS/dRI. The error bars indicate standard deviation of the determined molecular weights given by Debye fitting. (**B**) M_w_ of the injection series in correlation to the dilution factor (concentration determined by dRI/concentration before injection). Dotted line indicates an exponential decrease fit of b and c. The two very first injections are marked in red.

**Figure 10 pharmaceutics-12-01143-f010:**
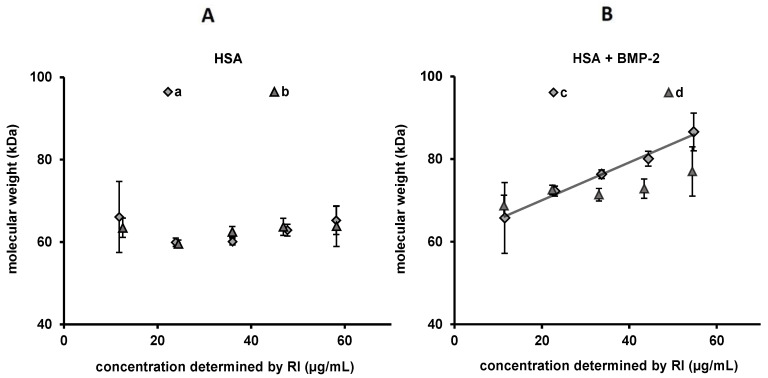
Molar masses of two sets of measurements with a one-day interval in PBS buffer pH 7.4. (**A**) Diluted from a 1.15 mg/mL HSA solution. (**B**) Diluted from a mixture of 0.2 mg/mL BMP-2 and 1.15 mg/mL HSA. Each symbol represents a single measurement. Error bars indicate standard deviation of the determined molecular weights given by Debye fitting.

**Figure 11 pharmaceutics-12-01143-f011:**
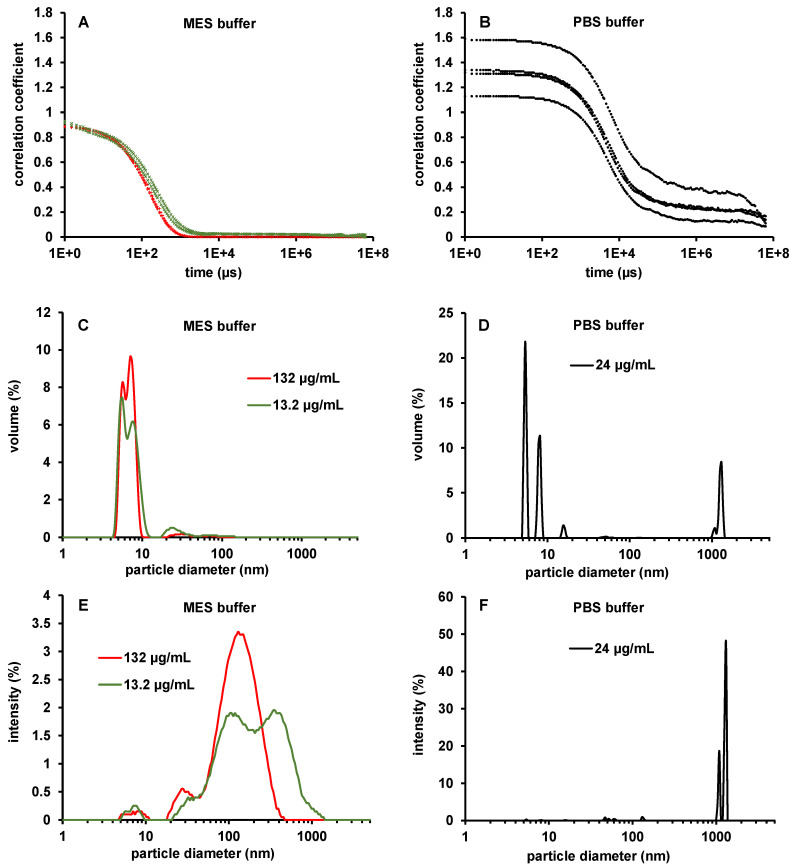
Autocorrelation functions of dynamic light scattering (DLS) measurements of BMP-2 in (**A**) MES buffer and (**B**) PBS buffer. (**C**) DLS size distribution of BMP-2 in MES buffer pH 5 by volume and (**E**) by intensity and (**D**) DLS size distribution of BMP-2 in PBS pH 7.2 by volume and (**F**) by intensity.

**Figure 12 pharmaceutics-12-01143-f012:**
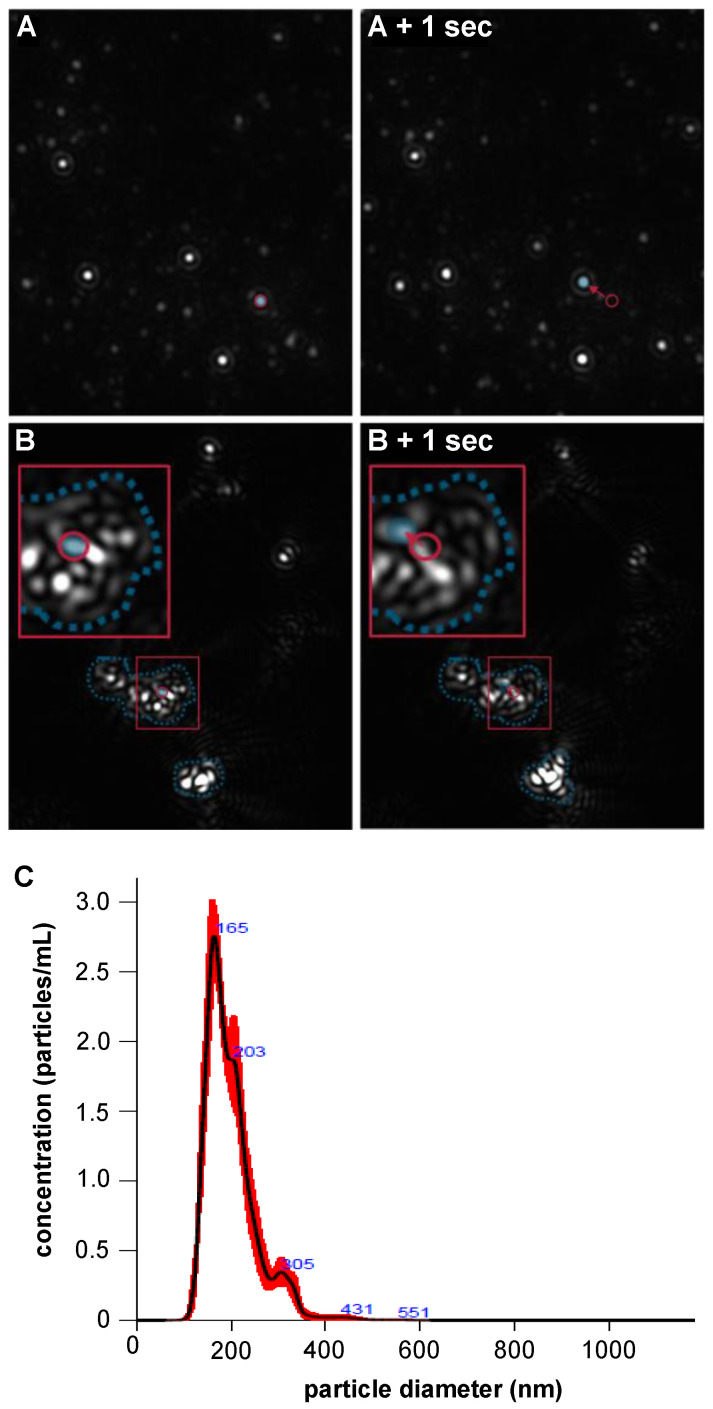
Single frames from microscopic light scattering videos (with same magnification) of samples containing 13.2 µg/mL BMP-2 during NTA in (**A**) MES buffer pH 5 and (**B**) PBS buffer pH 7.2. Red circles indicate individual intensity tracks. Blue dotted lines show large aggregates. (**C**) Mean NTA number size distribution (*n* = 5) of BMP-2 aggregates in MES buffer pH 5.

**Figure 13 pharmaceutics-12-01143-f013:**
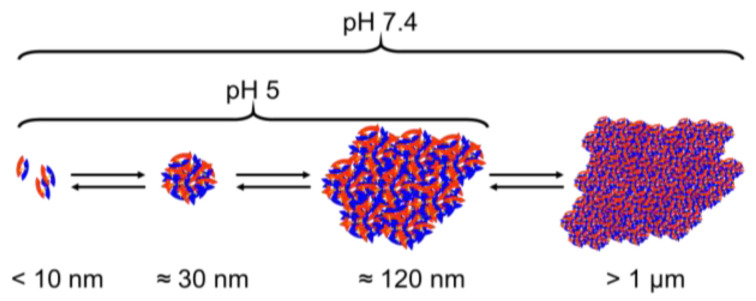
Schematic presentation of the apparent aggregation equilibrium of BMP-2 at pH 5 and pH 7.4.

**Table 1 pharmaceutics-12-01143-t001:** Solutions used in AF4/MALS experiments (cf. 3.4).

BMP-2 (mg/mL)	HSA (mg/mL)	Buffer
0.452	-	MES
0.226	-	MES
0.226	-	PBS
-	1.15	PBS
0.200	1.15	PBS

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
