# Peer review of "Bone Morphogenetic Protein 2 (BMP-2) Aggregates Can be Solubilized by Albumin—Investigation of BMP-2 Aggregation by Light Scattering and Electrophoresis"

_pharmaceutics, 2020, doi:10.3390/pharmaceutics12121143_

Round 1

Reviewer 1 Report

This is a very interesting work about the aggregation process of BMP-2 and the influence of the interaction between BMP-2 and Human serum albumin protein in this process. The article is clear and well write and provides interesting information to try to avoid these aggregation processes by means appropriate Pharmaceutical formulation. Therefore I consider that it can be accepted for publication in Pharmaceutics

Reviewer 2 Report

In this well-written manuscript, Sundermann et al., presents a study that elucidates the effects of albumin on bone morphogenetic protein 2 (BMP-2) aggregation and solubility. A member of transforming growth factor-beta, BMP-2 induces bone formation and has been investigated for medical purposes; however, BMP-2’s high tendency to aggregate at physiological pH and ionic strength complicates its delivery. Mapping the electrostatic potential reveals a tripolar charge distribution, and two BMP-2 molecules can associate in manners that complement the charged and hydrophobic regions. Combining light scattering and electrophoresis techniques, the authors demonstrated that large yet reversible BMP-2 aggregates were concentration- and pH-dependent, and that binding with albumin formed negatively charged complexes that solubilized BMP-2 aggregates. Findings from this study will have a significant impact on pharmaceutical research and treatment strategies for bone growth and healing.

The following are this reviewer’s comments/suggestions/questions regarding the results/interpretations reported and discussed in the manuscript that the authors may want to address:

  1. This reviewer finds Figure 1 informative as the diagrams detail the principle behind the 2D gels, which sets a solid foundation for readers to understand the study.
  2. (Minor comment) Albumin is a promiscuous binder. Potential off-target binding can occur in the body, so how safe will the new albumin formulation be? Have there been previous FDA-approved medications that used albumin to improve solubility?
  3. (Minor comment) The introduction can benefit from an explanation that elaborates on why albumin was selected to solubilize BMP-2. Furthermore, was albumin acting as a molecular crowder or a ligand? What is the affinity of albumin-BMP-2 binding? Is the complex functionally active?
  4. (Minor comment) This manuscript mentioned the effects of salt concentration, pH, and ligand binding on BMP-2 solubility. This reviewer suggests that the authors might want to investigate osmolytes as potential agents in improving BMP-2 solubility.
  5. In Figure 9 figure legend, symbols for the left graph can be combined with the titles on the right graph. Also, addition of colors will improve legibility.
  6. While Figure 13 already provides a diagram that effectively summarizes the impact of pH on reversible BMP-2 aggregates, this reviewer believes it will be even more informative to include albumin and its effects on the aggregation process.
  7. In Figure 10, HSA + BMP-2 molar mass measurements have significant error bars overlap. Additional measurements with averaging are needed to resolve the error.
  8. In Figure 11, experiment in PBS buffer should be done with protein concentrations similar to those in MES buffer to aid interpretation. An additional explanation should be included in the result or discussion to clarify if those concentrations are unfeasible.
